# Predicting the potential distribution of *Dactylorhiza hatagirea* (D. Don) Soo-an important medicinal orchid in the West Himalaya, under multiple climate change scenarios

Laxman Singh[1], Nidhi Kanwar[2], Indra D. Bhatt◉[1]*, Shyamal K. Nandi[1], Anil K. Bisht[3]

1 Center for Biodiversity Conservation and Management, G. B. Pant National Institute of Himalayan Environment, Kosi-Katarmal, Almora, Uttarakhand, India, 2 Center for Environment Assessment and Climate Change, G. B. Pant National Institute of Himalayan Environment, Kosi-Katarmal, Almora, Uttarakhand, India, 3 Department of Botany, D. S. B. Campus, Kumaun University, Nainital, Uttarakhand, India

* id_bhatt@yahoo.com, idbhatt@gbpihed.nic.in

**Data Availability Statement:** All relevant data are in the paper and supporting information files.

## Abstract

Climate variability coupled with anthropogenic pressures is the most critical driver in the Himalayan region for forest ecosystem vulnerability. *Dactylorhiza hatagirea* (D.Don) Soo is an important yet highly threatened medicinal orchid from the Himalayan region. Poor regenerative power and growing demand have resulted in the steep decline of its natural habitats populations. The present study aims to identify the habitat suitability of *D. hatagirea* in the Western Himalaya using the maximum entropy model (MaxEnt). The community climate system model (CCSM ver. 4) based on representative concentration pathways (RCPs) was used to determine suitable future areas. Sixteen least correlated (< 0.8) bioclimatic, topographical and geomorphic variables were used to construct the species climatic niche. The dominant contributing variables were elevation (34.85%) followed by precipitation of the coldest quarter (23.04%), soil type (8.77%), land use land cover (8.26%), mean annual temperature (5.51%), and temperature seasonality (5.11%). Compared to the present distribution, habitat suitability under future projection, i.e., RCP 4.5 and RCP 8.5 (2050 and 2070), was found to shift to higher elevation towards the northwest direction, while lower altitudes will invariably be less suitable. Further, as compared to the current distribution, the climatic niche space of the species is expected to expand in between 11.41–22.13% in the near future. High habitats suitability areas are mainly concentrated in the forest range like Dharchula and Munsyari range, Pindar valley, Kedarnath Wildlife Sanctuary, West of Nanda Devi Biosphere Reserve, and Uttarkashi forest division. The present study delineated the fundamental niche baseline map of *D. hatagirea* in the Western Himalayas and highlighted regions/areas where conservation and management strategies should be intensified in the next 50 years. In addition, as the species is commercially exploited illegally, the information gathered is essential for conservationists and planners who protect the species at the regional levels.

**Funding:** No funding available of this work

**Competing interests:** No authors have competing interest

## 1. Introduction

Climate is an important ecological and abiotic factor affecting species potential geographic distribution and ecological niche space [1]. The minimal changes in species bioclimatic envelope are thought to have considerable impacts on the plant-pollinator relationship, seed set, and regeneration status. It is expected that species may no longer adapt to a set of environmental conditions to facilitate further expansion [2]. Therefore, the species must either cope with the prevailing ecological conditions or colonize to sustain or become extinct [3]. This has led to a growing interest in developing and scaling up prioritization strategies for such species to ensure the highest conservation gains [4].

The Himalayan regions are an assemblage of biodiversity hotspots [5]. However, the ongoing disturbance exacerbated by climate change, habitat fragmentation, invasions by alien species, grazing and trampling, overexploitation, and excessive consumption of natural resources has altered the structural and functional integrity of the various Himalayan ecosystems [6]. Besides these, climate variability, land-use change, and rural migration are key contributors to biodiversity loss in the region [7]. In the last few decades, the region saw cascading effects of climate variability mainly due to increased greenhouse gases concentration. It is believed that the rate of global warming in the Himalayas is much higher than the global average. For instance, in the last 100 years, the global average temperature rise was 0.74˚C [8]. However, in the Himalayas, a 1.5˚C temperature increase was documented in the final quarter (i.e., 1982–2006) of the twentieth century [9], with warming potentially reaching 5˚C by the end of the twenty-first century [9, 10]. This rise is alarming because Himalayan floras are alienated to specific elevation gradient/microhabitat conditions [3, 11]. The shift in the climatic envelope is expected to bring significant change in the resident species habitat conditions, leading to changes in species richness, population structure, and those unable to cope are likely to face local extinction [1–3].

Furthermore, the recent upsurge in herbal or its derived products across the globe has led to uncontrolled abusive practice; thus, the natural stock of these plants is under tremendous pressure. In the case of the Indian Himalayan Region (IHR), a considerable number (1748 species) of medicinal and aromatic plants (MAPs) are reported, with 31% of them being native, whereas 15.5% are endemic and threatened [12]. The high potential instability and inherent vulnerability make the region one of the most ecologically fragile bio-geographic zones [13]. Other challenges on the MAPs include low population size, habitat specificity, genetic bottleneck effect, narrow distributional ranges, and heavy livestock grazing [14]. The literature on these threatened plants is fragmentary or limited to specific geographic pockets [15]. In the above context, it is obligatory to make a conservation framework encircling species habitat restoration and promote cultivation, thus, reducing pressure on the wild populations.

The development of statistical modelling and geospatial technology in predicting suitable habitat distribution has gained immense popularity. However, such information is at an initial phase for the Himalayan MAPs [6]. The use of geospatial technology could add an advantage as obtaining specific distribution maps for such species is difficult and often requires intensive surveys [16]. The difficulty level becomes amplified in the Himalayan region where the working conditions are not conducive for the survey, i.e., inaccessible and difficult topography perplexed with hostile conditions. Therefore, estimating current plants distribution and identifying important climatic refugia will help predict future distribution patterns and reveal regions with high extinction rates. At present, the common method to study potentially species distribution and environmentally suitable habitats is to use species distribution models (SDMs) [17]. SDM has made it possible to analyze the environmental drivers of species

distributions and project a species realized niche into a geographic area [18]. Of many SDM algorithm methods, MaxEnt has proved decisive when modeling rare species with narrow ranges [19]. MaxEnt modelling is a robust computational algorithm that works on the backdrop of species presence points and rasterized environmental data. The probable ecological niche can be reconstructed using species presence data points and environmental variables/ predictors [20]. Such model-based sampling would become an important benchmark for endemics and threatened species and is a well-recognized cost-efficient method [21].

In the present study, an effort has been made to model the potential habitat distribution and effect of future climate change on *D. hatagirea*, a critically endangered [22, 23] and endemic species of the Himalayan region. The species is a tall, slender, ground-dwelling, perennial herb with palmately lobed tuberoids that prefer to grow in a moist, mild, acidic soil environment (Fig 1A and 1B). The species has been reported from India, Afghanistan, Pakistan, Nepal, Tibet, and Bhutan [6]. In the IHR, the species is reported from Jammu and Kashmir, Ladakh, Himachal Pradesh, Uttarakhand, Sikkim, and Arunachal Pradesh at an altitudinal range of 2500 to 4500 m above sea level (asl). The estimated annual trade of the species is around 10–50 metric tons [24], with an economic value of US $ 68.88–89.54 (1US $ = Rs. 77.39) kg$^{-1}$ of the dried tuber (Fig 1C). At present, the tuber of the species is destructively harvested and illegally traded; thus, it puts a stake on its future existence. Moreover, the species require specific microhabitat conditions for growth and perturbation, thus limiting the species widespread distribution. Therefore, to minimize the pressure on the wild populations, efforts are ongoing to develop and upscale the existing multiplication strategies for mass multiplication. Meanwhile, mapping and conserving the critical habitat is expected to offer a possible solution to species conservation and management. The study attempts to address the following scientific questions: (i) What is the present potential geographical

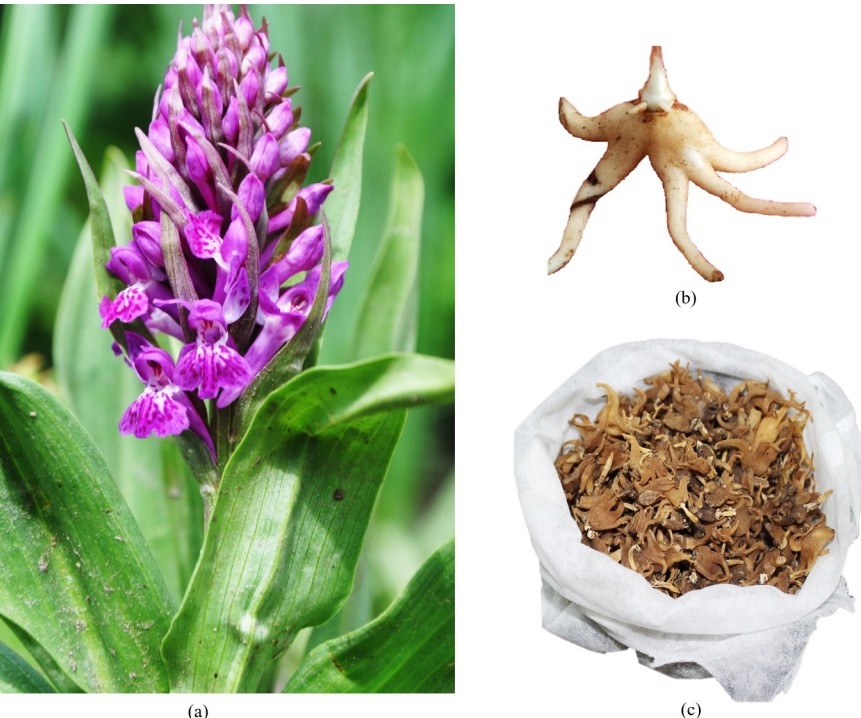

(a)

(b)

(c)

**Fig 1.** Image showing (a) compact floral structure of *D. hatagirea*, (b) well developed palmately lobed tuber, and (c) tuber collected and processed as a marketed product.

distribution range of *Dactylorhiza hatagirea* in the Western Himalaya, India? (ii) What will be the impacts of climate variability on the future distribution of *D. hatagirea* using four Representative Concentration Pathways (RCPs)?, and (iii) Where are the high potential distributional areas of *D. hatagirea* that could be protected or could be suggested for cultivation, reintroduction/ recovery plans?. Answering these questions will help identify suitable habitats for the conservation of the species, which may help policy planers while developing strategies for its conservation.

## 2. Materials and methods

### 2.1. Study area and ecological significance

The study was undertaken in Uttarakhand state (28˚43′ to 31˚28′ N Latitude and 77˚34′ to 80˚03′ E Longitude) of IHR (Fig 2). The state has a total recorded forest area (RFA) of 38,000 km$^2$ (71.05% of its total geographical area 53,485 km$^2$), out of which 26,547 km$^2$ is reserved

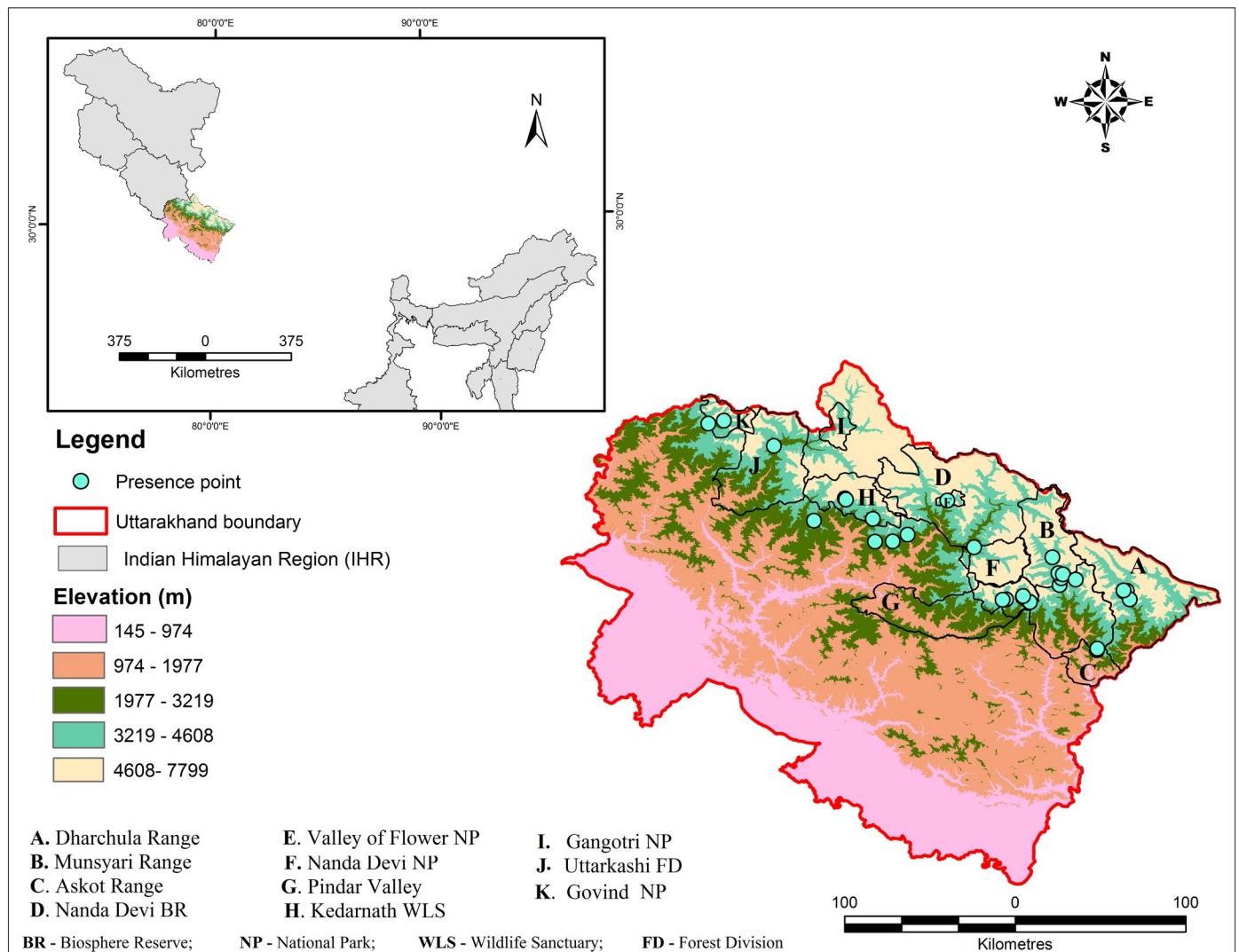

**Fig 2. Field collected *D. hatagirea* (blue dots) points mounted on the elevation map of Uttarakhand state.** Maps in Fig 1 are generated with ArcGIS version 10.3 (ESRI, CA, USA).

forest, 9,885 km$^2$ is protected forest, and 1,568 km$^2$ is unclassed forests [25]. The state experience varied climates from warm dry to warm wet and with a latent cool, dry period. The state's temperature ranges from sub-zero to 43°C [26], and average annual rainfall varies from 1093.8 mm to 1385.5 mm [27].

The present study area harbors alpine vegetation, which covers 8,524 km$^2$. Of these, 4,376 km$^2$ is surmounted by permanent snow cover (i.e., corresponding to ca.24.11% statealpine geographical area) [28]. The alpine regions are well known for their high-value MAPs, including *D. hatagirea*. The region is experiencing major environmental transformation repercussions, and anthropogenic activities outnumber the natural eventualities, thus enforcing species to various threat categories [15].

## 2.2. Species point data

To predict species distribution, it is a prerequisite to have species presence points and environmental variables [29]. The data search was primarily made from online portals such as Global Biodiversity Information Facility [30], published literature, and herbarium consultation Botanical Survey of India, Dehradun (BSID). Data on the species were very limited, whereas herbarium records were not geo-referenced. Considering these limitations, an extensive field survey was conducted during 2016–19, and presence points were recorded. A total of 30 occurrences of the species were recorded during field surveys. A portable multi-channel Global Positioning System (Garmin) receiver with 10–20 m positional accuracy was used to record the species occurrence geo-coordinates. The coordinates were then converted to decimal degrees and used to model the species potential habitats distribution in its native range.

## 2.3. Data source

The climate data were downloaded from World Climate Database [31]. WorldClim provides current (baseline) and projected climate data for 2070 with a spatial resolution of 30 seconds (ca. 1 km) in GeoTIFF format. These climatic data are derivatives of maximum, minimum, and average values of monthly, quarterly, and annual temperatures and precipitation of the last 30 years, i.e., 1970–2000. Likewise, environmental variables such as soil type, soil moisture, and soil pH were downloaded from the International Soil Reference Information Centre [32], while land use land cover (LULC) from http://www.esa-landcover-cci.org/ [33] (Table 1). Besides these, non-climatic variables, i.e., altitude, aspect derived from NASA Shuttle Radar Topographic Mission (SRTM, version 4.1) [34]. The reason behind using both the climatic and non-climatic variables is to enhance the model's predictive power as suggested for endemic plants [5, 35]. Further, for future prediction studies, we used Community Climate System Model (CCSM) ver. 4 (CCSM4) that is based on the Fifth Assessment Report (AR 5) of the Intergovernmental Panel on Climate Change [36] and two contrasting Representative Concentrations Pathways (RCP 4.5 and RCP8.5) for the years 2050 and 2070. Furthermore, we assumed that the edaphic properties are expected to remain stable in the next several decades, as soil properties should not change synchronously with sudden climate change; hence the same raster layer was used in future projections.

**2.3.1. Environmental layers and variable selection.** The model's output can be accurate, biologically meaningful, and generalized if built with predictor variables that directly impact species distribution. Strong collinearity between the variables in SDMs may cause model overfitting due to the high level of correlation among variables [37]. To avoid multi-collinearity among the 19 bioclimatic variables, highly correlated variables (r ≥ 0.80 Pearson correlation coefficient) were eliminated from further models using ENM Tools. This reduction of predictor variables resulted in the inclusion of nine bioclimatic variables and seven environmental

**Table 1. Environmental variables and their percent contributions for predicting the potential distribution of *D. hatagirea*.**

| Type | Code | Variable name | Unit | % Contribution |
|---|---|---|---|---|
| Climatic | Bio1 | Annual mean temperature | ˚C | 5.51 |
| | Bio2 | Mean diurnal range (mean of monthly max. and min. temp.) | ˚C | 0.07 |
| | Bio3 | Isothermality [(Bio2/Bio7) x 100] | - | 0.98 |
| | Bio4 | Temperature seasonality (standard deviation x 100) | C of V | 5.11 |
| | Bio7 | Temperature annual range (Bio5-Bio6) | ˚C | 1.48 |
| | Bio8 | Mean temperature of wettest quarter | ˚C | 0.08 |
| | Bio12 | Annual precipitation | mm | 1.95 |
| | Bio18 | Precipitation of warmest quarter | mm | 4.38 |
| | Bio19 | Precipitation of coldest quarter | mm | 23.04 |
| Geomorphologic | DEM | DEM | ° | 34.85 |
| | SLP | SLOPE | ° | 3.06 |
| | | ASPECT | | 0.98 |
| Pedologic | | SOIL TYPE | | 8.77 |
| | | SOIL MOISTURE | mm | 1.13 |
| | | SOIL pH | | 0.36 |
| Land use land cover | | LULC | | 8.26 |

variables for the prediction process (S1 Table). Further, using ArcGIS 10.3, all predictor variables layers were rasterized into the same bounds, cell sizes, and coordinate system as the layer of occurrence localities. Finally, these layers were converted to ASCII format for further processing in MaxEnt.

**2.3.2. Model parameterization.** MaxEnt algorithm (MaxEnt ver. 3.4.1) [38] for habitat distribution modelling was employed [29]. MaxEnt algorithm was chosen over other available machine learning tools owing to (i) presence of only data points of the species, (ii) works even relatively with a small number of occurrence locations and high predictive performance, (iii) can handle continuous and categorical environmental data simultaneously, (iv) analyze results in terms of percent contribution of environmental data through model output, (v) examine variables weight through jackknife method, and (vi) calibrate the model, run numerous replicates along with cross-validation, and bootstrapping to test model robustness [18, 39, 40]. We used 75% of the dataset for training and 25% dataset model testing in this study. For generating model robustness, the number of iterations was set to 5000, with 30 replicated model runs. The maximum background points10000 and ten percentile training presence with logistic threshold rule were applied, whereas other parameters were set to default.

**2.3.3. Model performance and potential niche change.** To calibrate the model and validate its robustness, threshold independent receiver-operating characteristic analysis (ROC) and area under the receiver-operating characteristic curve (AUC) were tested for model precision. The AUC value varies between 0 to1. The values close to +1 indicate conformity between observations and prediction, whereas zero or less values indicate a performance no better than random [41]. Statistically, AUC values near 1 indicate very good model performance, whereas AUC values close to 0 signify complete inaccurate prediction. Model performance based on AUC values are categorized as, very good (0.95 < AUC < 1.0), good (0.9 < AUC < 0.95), fair (0.8 < AUC < 0.9), and poor (AUC < 0.8) [42, 43]. In the past, several studies have suggested that the AUC values mislead the performance of predictive distribution models and reflect relative model performance [44]. Therefore, to assess the predictive success of models, sensitivity, specificity, overall accuracy, and True Skill Statistics (TSS) were calculated by a confusion matrix. Threshold-dependent TSS is considered an additional accuracy measurement that is

not affected by prevalence as it does for the kappa coefficient and the size of the validation set [45]. It deals with sensitivity and specificity, values ranging from − 1 to + 1, where + 1 indicates perfect agreement, scores ranging from 0.6 to 0.9 specify fair to good model performance, and 0 represents a random fit [45]. For this, the output of the logistic layer derived from MaxEnt results was reclassified into a binary prediction map (unsuitable and suitable) with a threshold of 10 percentile training presence. All geographical plotting and suitable range-size estimation were conducted in ArcGIS software (version 10.3).

To identify the potential area of distribution, the distributional indices based on threshold interval classification (TIC) were categorized as highly suitable (TIC > 0.75), moderate suitable (0.50 < TIC < 0.75), least suitable (0.25 < TIC < 0.50), and unsuitable areas (TIC < 0.25). Changes in the potential niche of *D. hatagirea* between the current and future climatic scenarios were computed by converting ASCII output projections into raster format using ArcGIS 10.3. Simultaneously the number of cells (pixels) among the projected climatic extent was calculated using zonal statistics in spatial analyst tools in ArcGIS 10.3. The differences in the mean number of cells among four classes of potential habitats were converted to surface area (km$^2$). Finally, MaxEnt predictive maps for the current and future scenarios were related to elevation classes. This would help map habitats and contribute to species-specific interventions/ reintroduction programs.

# 3. Results

## 3.1. Preliminary screening of model inputs variables

The credibility of any prediction model is dependent on input variables for species distribution modelling. Given this, sixteen predictor variables out of twenty-six variables; with correlation coefficients of ≤ 0.8 were retained after preliminary screening and selected for further modelling (Table 1).

## 3.2. Model performance and variable contributions

The results obtained by an ecological model are judged for their performance based on complex algorithm tests and model validation. The threshold-independent ROC showed that the average AUC yielded satisfactory results of 0.96 (Fig 3), which falls under 'very good' (0.95 < AUC < 1.0) model performance based on Thuiller et al. (2005) [42] classification. The confusion matrices for the current prediction model calculated the model's sensitivity and specificity to be 0.79 and 0.95, respectively. With these matrices in place, the model performance (i.e., TSS) was calculated (sensitivity + specificity—1). The TSS for the current model was computed to be 0.74, which indicates that the model's overall performance was good, based on Allouche et al. (2006) [45] criteria.

The variable contributions analysis highlights; elevation had the most (34.85%) influential effect followed by precipitation of coldest quarter (Bio 19; 23.04%), soil type (8.77%), LULC (8.26%), annual mean temperature (Bio 1; 5.51%), temperature seasonality (Bio 4; 5.11%), precipitation of warmest quarter (Bio 18; 4.38%) and Slope (3.06%) (Table 1). The variables mentioned above cumulative contributions stood at ~93% to the modeled potential climatic niche of *D. hatagirea*. Similarly, Jackknife analysis indicates annual mean temperature (Bio 1), elevation, mean temperature of the wettest quarter (Bio 8), and precipitation of coldest quarter (Bio 19) as most important predictor variables (Fig 4). These variables provide useful and distinctive information defining the *D. hatagirea* distributions when used in isolation. Variables like soil type, LULC, temperature seasonality (Bio 4), and annual temperature range (Bio 7) showed considerable change and showed moderate gain when used separately (Fig 4). Furthermore, the quantitative relationship between

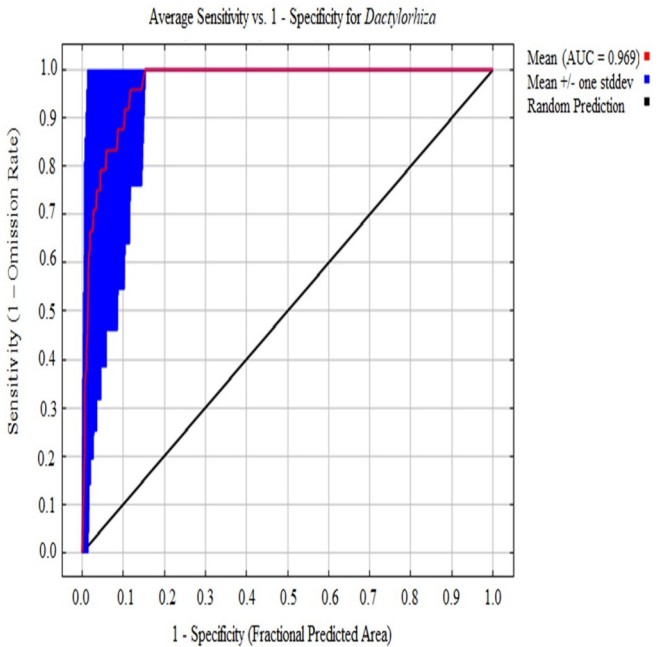

**Fig 3. Receiver operating characteristic curve with area under the curve (AUC) signifying model robustness.**

the logistic probability and input variables are depicted as response curves (Fig 5). A geo-morphic variable, such as elevation, was one of the key variables that describe the present and future spatial distributions of *D. hatagirea*. Response curves analysis reveals average altitude ranged from 2800 m to 4500 m, precipitation of coldest quarter (Bio 19) ranged from (150 mm to 380 mm), annual mean temperature (Bio 1) ranged in between (0 – 25˚C). Likewise, precipitation of the warmest quarter (Bio 18) ranged in between (200 mm-1100 mm), soil pH (5–6), and slope angle ranged from (5˚ - 45˚). Thus, all the identified variables estimate the important climatic attributes that potentially influence the distribution of *D. hatagirea* in northwestern Himalaya, India.

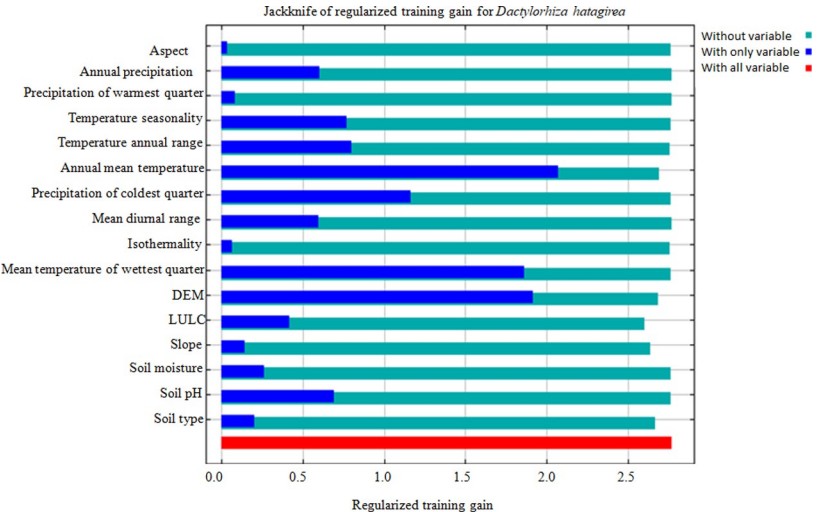

**Fig 4. Results of jackknife test highlighting the relative importance of each variable when used in isolation.**

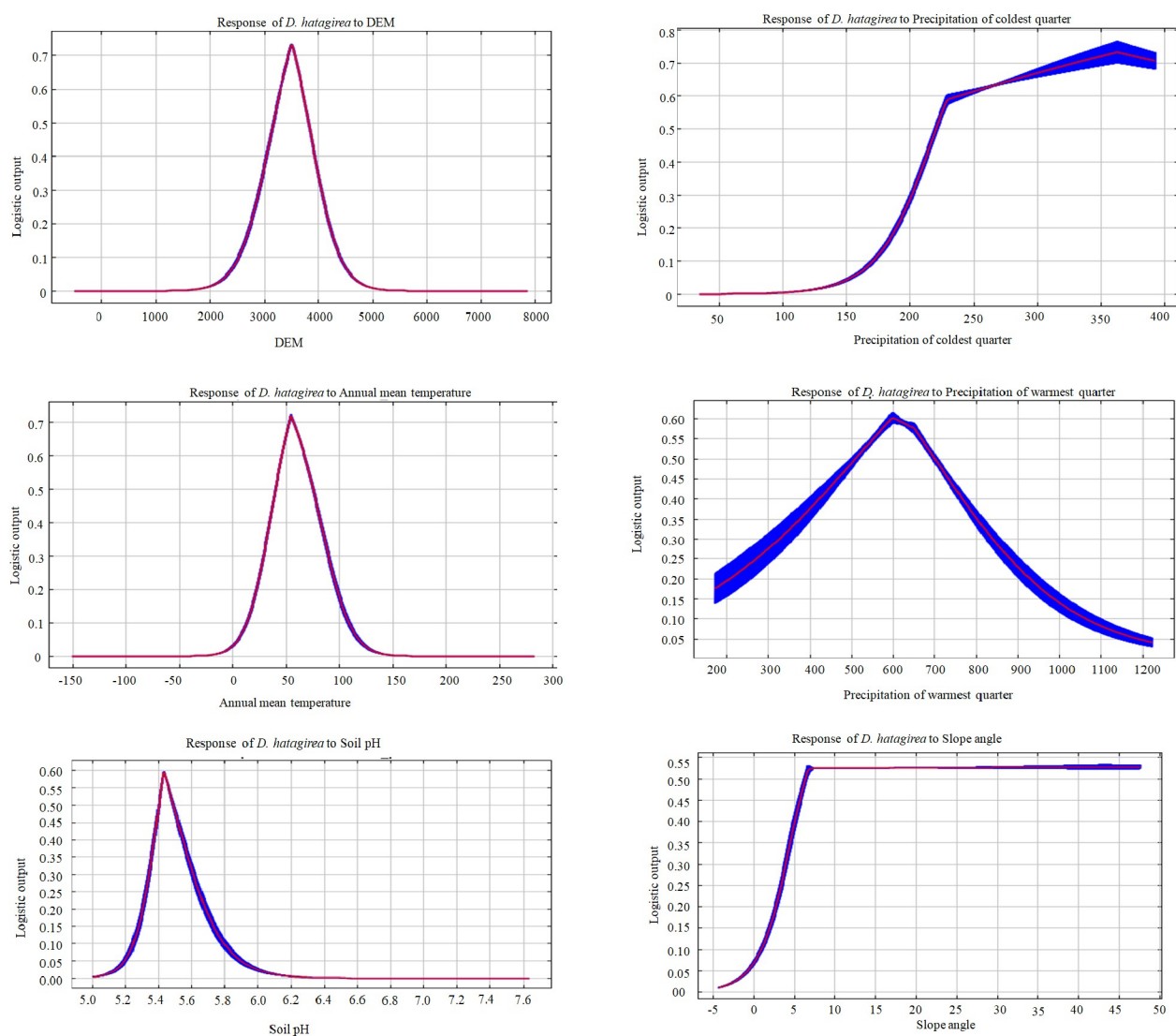

**Fig 5. Response curves of six environmental predictors and their relationships with the probability of the target species suitability range.**

### 3.3. Current predicted potential distribution of climatically suitable areas

Habitat suitability of *D. hatagirea* was determined based on threshold interval classification (TIC). Of these, maximum of 96.18% (51637 sq. km) of the geographic region was predicted to

**Table 2. Predicted habitat suitability area (km²) of *D. hatagirea*, in the present and future climate change scenarios (RCP 4.5–8.5; 2050 and 2070).**

|  |  | Present | RCP 4.5 2050 | RCP 8.5 2050 | RCP 4.5 2070 | RCP8.52070 |
|---|---|---|---|---|---|---|
| **Range** |  | Area (km²) | | | | |
| 0.00–0.25 | Not Suitable | 51637 | 51478 | 51526 | 51513 | 51564 |
| 0.25–0.50 | Low Suitable | 1664 | 1777 | 1734 | 1729 | 1684 |
| 0.50–0.75 | Moderately Suitable | 255 | 279 | 281 | 285 | 292 |
| 0.75–1.00 | Highly Suitable | 131 | 153 | 146 | 160 | 147 |
| Total |  | 53687 | 53687 | 53687 | 53687 | 53687 |

be unsuitable (TIC < 0.25), followed by 3.09% (1664 sq. km) with least habitat suitability (0.25 < TIC < 0.50), and moderate suitability being 0.47% (255 sq. km). The potential habitat with high suitability accounts for only 0.24% (131 sq. km) of the state's total geographic area (Table 2). The areas with high habitats suitability (TIC > 0.75) are mainly concentrated in the forest range like Dharchula and Munsyari range, Pindar valley, Kedarnath Wildlife Sanctuary, West of Nanda Devi Biosphere reserve, and Uttarkashi forest division (Fig 6). Likewise, moderate habitat suitability was located in GovindVihar National Park, Uttarkashi forest division, Kedarnath Wildlife Sanctuary, Nanda Devi Biosphere Reserve, and National Park, Pindar valley and in the forest range of Dharchula and Munsyari range.

### 3.4. Future projection of climatically suitable areas of *D. hatagirea* distribution

Future projection habitat suitability map under the CCSM4 model for RCP 4.5 and RCP 8.5 (2050 and 2070) is very similar to the current distribution (Table 2). The present study results depict the geographic distribution of the species would expand under predicted levels of climate change (RCP 4.5 and RCP 8.5) compared with the current potential distribution (Figs 6 and 7). High habitat suitability under the RCP 4.5 scenario; predicts an increase of 0.4% (22.10 sq. km) for 2050 and 0.3% (15 sq. km) for 2070. Under the RCP 8.5 projection, an increase of 0.5% (29 sq. km) is expected in 2050 and 0.27% in 2070. Although, the potential high suitability increases under both the scenarios (RCP 4.5 and 8.5) when compared with the current prediction, the rate of increase for the year 2050 is comparatively higher, after which (towards 2070) it showed a decreasing trend.

### 3.5. Shifts in habitat suitability under the climate change scenarios

The final output maps (current and future scenarios) were employed to find out habitats that will remain stable, gains in habitat area, habitat loss, and unsuitable habitat as part of computing change analysis (in sq. km) (Table 3) (S1 and S2 Figs). The change analysis highlights; only 1992 sq. km of stable habitat (maximum) under RCP 4.5 (2050) then under RCP 8.5 (2070) with a minimum stable habitat of 1928 sq. km. Stable habitats showed a decreasing trend with the climate projection model. Likewise, under RCP 4.5 (2050), an area of 107 sq. km was recorded as a gain in habitat, followed by 102 sq. km in 2050 (RCP 8.5) and 93 sq. km (RCP 8.5) in 2070. Habitat gain indicates the region becomes more suitable for the species under future climatic conditions. Besides these, habitat losses (contraction) were recorded. Of these, maximum (112 sq. km) contraction was seen in 2070 (RCP 4.5), followed by 110 sq. km in 2050 (RCP 4.5) and 105 sq. km in 2050 (RCP 8.5) (Table 3). The contraction in habitats was mainly recorded from the low habitat suitability class. A detailed shift in degrees in a different class is tabulated in Table 4.

## 4. Discussion

The susceptibility to climate change is now reflected in spatial distribution and forest ecosystem vulnerability across the globe [46]. Climate change is projected to be a 'dominant stressor' under different climate projection models in the latter half of the 21st century [47]. As there is no denial of the reality of climate change, attention is now being actively paid to formulate some mitigation measures to maintain the stability of an ecosystem. Species distribution models in this direction have contributed immensely by providing reliable information about the potentially suitable habitats of sensitive/ vulnerable species or communities that need priority attention. The present study investigates the habitat suitability of *D. hatagirea* under a changing climate scenario using a species distribution model. The study provides a detailed map of the current and future species distribution in light of climate changes (Figs 6 and 7). *Dactylorhiza*

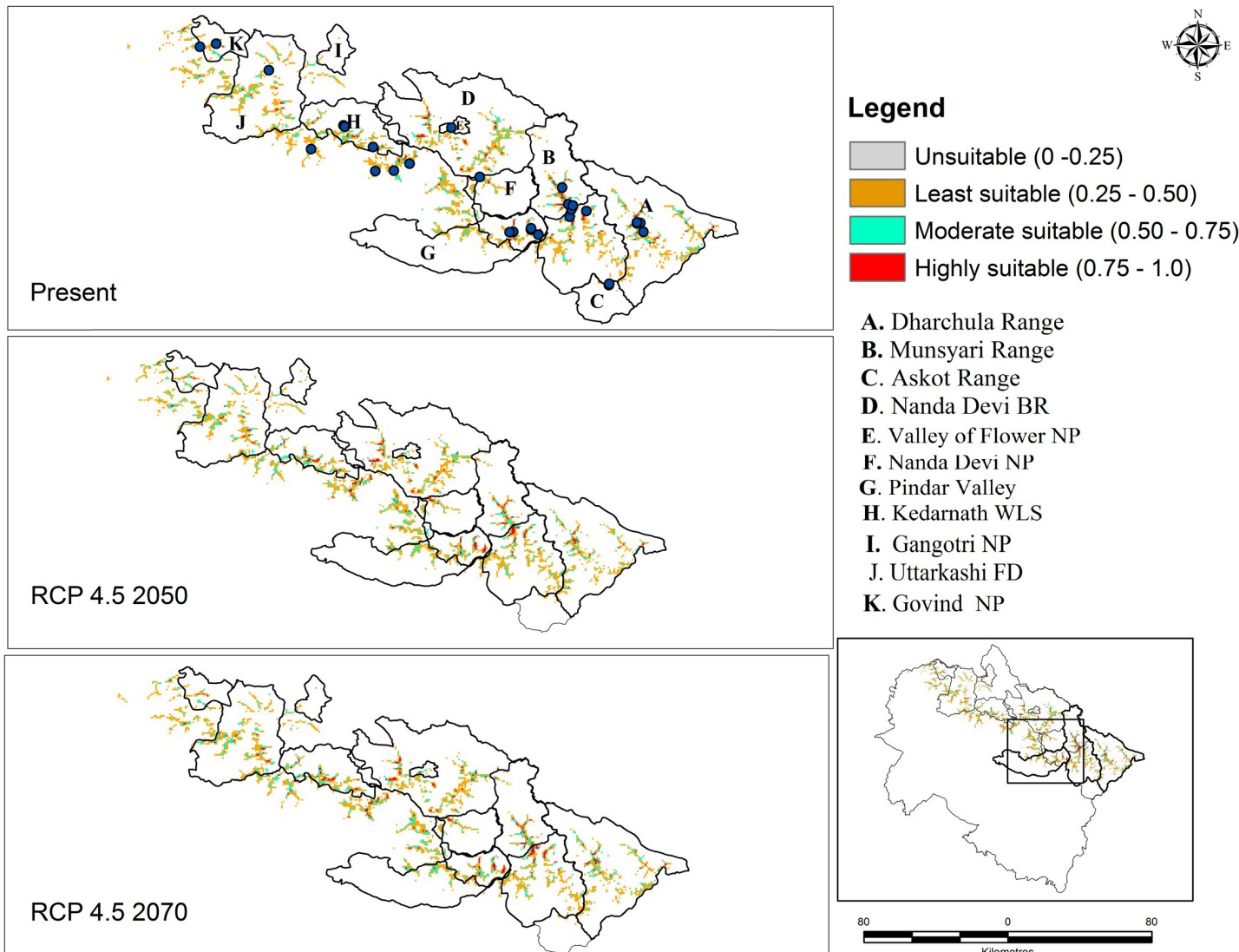

**Fig 6. Potential geographic distributions of current and under RCP 4.5 (2050 and 2070) of *D. hatagirea* in the Uttarakhand state.** Maps in Fig 6 are generated with ArcGIS version 10.3 (ESRI, CA, USA).

*hatagirea* distribution was mostly explained by topographical variables rather than bioclimatic variables (Table 1). Among the topographical factors, elevation (34.85%) was the most dominant contributing factor, followed by soil type (8.77%) and LULC (8.26%), while among the bioclimatic variables, precipitation of coldest quarter (Bio 19; 23.04%), annual mean temperature (Bio 1; 5.51%), and temperature seasonality (Bio 4; 5.11%) were most prevalent. These physiographic factors, along with topographical features (i.e., elevation, land use characteristics, slope angle, and aspect), and bioclimatic parameters, are reported to have pronounced effects on the pattern of species distribution and community structure in the alpine meadows [35, 48]. Such factors play a key role in the alpine biodiversity where the species are skewed more towards particular habitats (i.e., moist and marshy habitat) than open grasslands or rugged terrains [14, 49].

Previous findings are in line with this study, wherein altitude and bioclimatic variables (temperature and precipitation) were reported to play a major role in the distribution and population structure of *D. hatagirea* [50–53]. Among these, Thakur et al. (2021) [52] reported

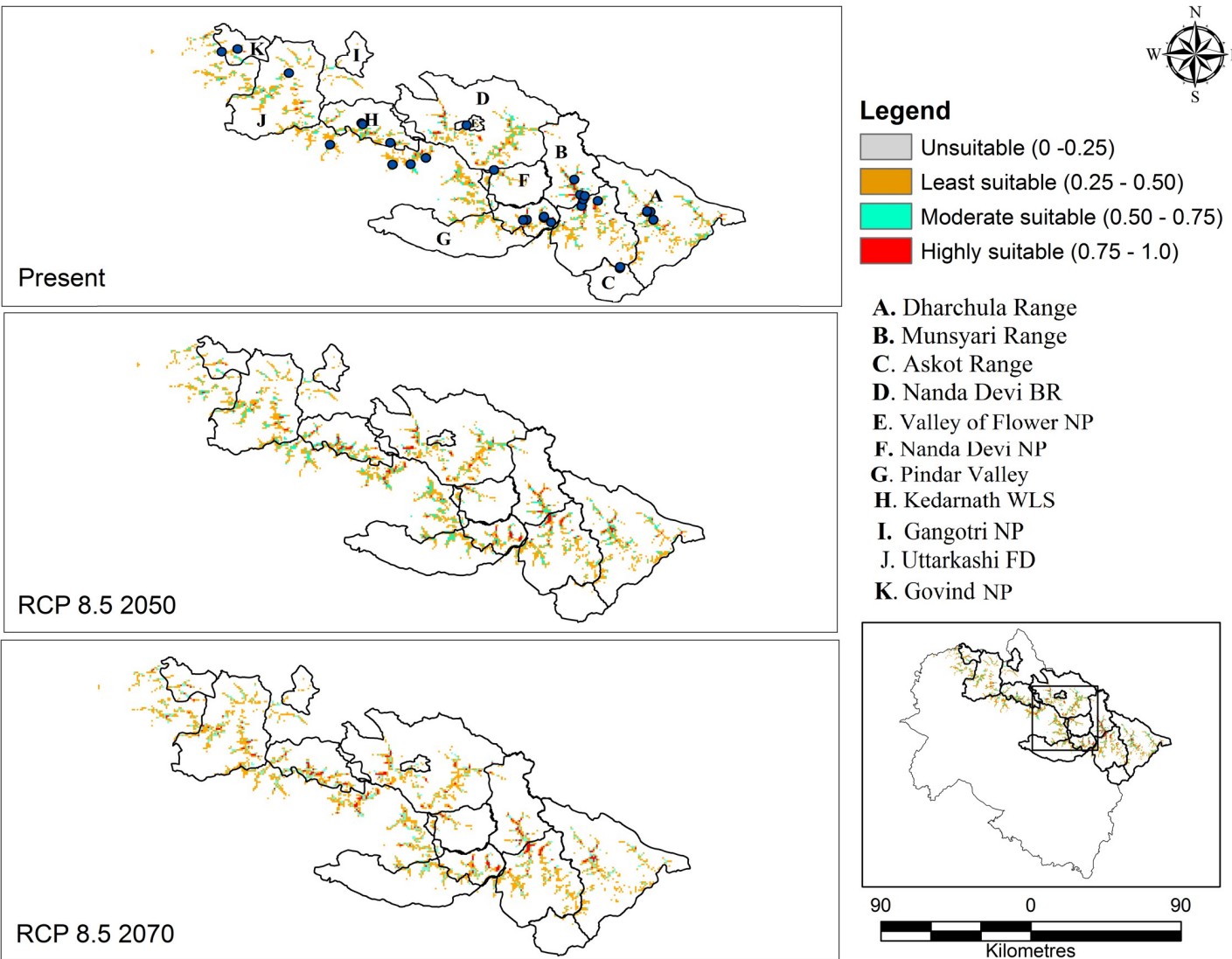

**Fig 7. n-depth view of predicted future habitat for *D. hatagirea* under RCP 8.5 (2050 & 2070). I** Maps in Fig 6 are generated with ArcGIS version 10.3 (ESRI, CA, USA).

precipitation of the coldest quarter (Bio 19) as the most significant bioclimatic variable that influences *D. hatagirea* distribution as obtained in this study. The ecological relevance of Bio 19 in the target species distribution was accredited to its marshy habitat, for which

**Table 3. Habitat transformation (km$^2$) as predicted by comparison with present distribution using RCP 4.5–8.5 (2050 and 2070).**

|  | RCP 4.5 2050 | RCP 4.5 2070 | RCP 8.5 2050 | RCP 8.5 2070 |
|---|---|---|---|---|
| Gain | 107 | 102 | 98 | 93 |
| Loss | 110 | 105 | 112 | 103 |
| Stable | 1992 | 1966 | 1951 | 1928 |
| Unsuitable | 51478 | 51514 | 51526 | 51564 |
| Total area | 53687 | 53687 | 53687 | 53687 |

**Table 4. Change detection of suitability class by comparison with present distribution as computed in Km² under RCP 4.5 and RCP 8.5.**

| | RCP 4.5 2050 | RCP 4.5 2070 | RCP 8.5 2050 | RCP 8.5 2070 |
|---|---|---|---|---|
| **Suitability class** | Shift in the area (km²) | | | |
| Unsuitable | 51478.4 | 51526 | 51513.8 | 51563.9 |
| Unsuitable to low suitable | 1099 | 1086 | 1079 | 1065.5 |
| Moderate suitable | 142 | 137 | 135 | 136 |
| Low suitable | 362 | 359 | 360 | 352.5 |
| New low suitable | 106.8 | 98.1 | 102 | 92.5 |
| Contraction | 109.7 | 112 | 105.3 | 102.5 |
| Low to unsuitable | 75.8 | 65.5 | 60.5 | 57.5 |
| Low to moderate suitable | 137 | 126 | 131 | 128 |
| Moderate to low suitable | 23.5 | 13.4 | 21.5 | 13.4 |
| Moderate to highly suitable | 52.9 | 54 | 56.4 | 50.1 |
| Unsuitable to moderate suitable | 0 | 18 | 19 | 27.5 |
| Low to highly suitable | 59.2 | 62 | 59.6 | 63.4 |
| Unsuitable to highly suitable | 41 | 30 | 44 | 33.5 |
| Total area | 53687 | 53687 | 53687 | 53687 |

precipitation (in the form of snow) during the winter season could have a direct role in recharging groundwater and maintaining desired soil moisture [52, 54]. Blinova (2008) [55] and Sletvold et al. (2010) [56] have shown that temperature (of the growing season) and precipitation are responsible for the persistence of *Dactylorhiza* populations. Further, a study by Shrestha et al. (2021) [53] from Nepal postulated a different stance where annual mean temperature (Bio1), precipitation seasonality (Bio15), and annual precipitation (Bio12) are the most significant variables in the target species distribution, while, precipitation in the coldest quarter (Bio19), precipitation in the driest quarter (Bio17), and the environmental layer were of intermediate significance. Similar to our findings, Rana et al. (2020) [3], using ensemble species distribution modelling (eSDM), reported elevation (30.97%) as the major dominant contributing variable, while amongst the bioclimatic variables, the mean temperature of the wettest quarter (Bio 8; 24.69%) and annual precipitation (Bio12; 21.11%) were the key contributing variables. Given the understanding from the above and several other studies on terrestrial orchids, other than biotic elements, temperature (of the growing season) and precipitation strongly modulate or affect their distribution. Thus, any significant climate change can be envisaged to have a magnified impact on these species overall distribution and growth performance. An observational study in the laboratory conditions noted maintaining the live germplasm of the species at controlled temperature chambers, i.e., 20°C, 30°C and in glasshouse condition (>35°C), revealed better growth performance and flower development at 20°C only. In comparison, growth ceased after the initial leaf development stage and later perished in temperature above 35°C, suggesting species narrow temperature tolerance regime for growth and perpetuation. Similar findings are reported elsewhere, where an excessive rise in temperatures was reported to affect the vegetative growth flower bud differentiation negatively and preclude regeneration of *Paeonia delavayi* [57], *Camellia sinensis* [37], *Dalbergia cultrata* [58], *Rosa arabica* [59], *Hippophae salicifolia* [60], *Fritillaria cirrhosa* [61], *Rhododendron niveum* [62], and *Ilex khasiana* [43].

In addition to the above, soil type, moisture ratio, and soil pH [49] are the other vital decisive variables that posture or play a role in shaping the distribution of *D. hatagirea* (Table 3). In a study by Thakur et al. (2021) [52] revealed that the populations of *D. hatagirea* flourished in soils rich in organic matter (i.e., loamy sand to sandy loam) and had adequate soil moisture

(~ 63.23%). Also, Shrestha et al. (2021) [53] emphasized geological substrate and soil properties to be the key factors determining the distribution of the species. Similar observation was recorded during our ecological assessment surveys wherein the target species has luxuriant growth in moist sites along the hill slopes, dodged within narrow streams, and on open undulating meadows with high soil moisture [49]. Much of these habitats had slope steepness angles ranging from gentle (14˚) to moderate slope (41˚) angle, while north/ north-east aspects were most prevalent [49]. These variables were identified as factors of importance in our study, considering their role in posturing the vegetation reserve in the alpine habitats. Therefore, their role cannot be ignored in the Himalayas context, where the influence of the microclimatic conditions edaphic factors largely prevail [63].

In recent years, rapid climate changes in the Himalayas have resulted in distributional changes for a wide range of taxa [64]. The aftermath of these events has seriously impacted the geographical distribution of species, with reports of some species migrating to higher elevations [1, 57, 65–68]. The alpine species in the Himalayas are habitat-specific and have a narrow distributional range; therefore, they are more vulnerable to extinction. The disproportionate effects on these species migrating to high latitude or elevation are attributed to climate change [65]. Our prediction showed that the shift in suitable habitat distribution to high elevations would gradually become more significant using the climate projection model. Model predicted climatically suitable habitat would expand under the RCP 4.5 and RCP 8.5 climate scenario toward north/ northeast direction and would become invariably less suitable at the lower altitudes (Figs 6 and 7). Further, we speculate that the north/ northeastward shift observed in our study could also be attributed to; North-facing aspects in the mountains are associated with higher soil nutrient content, higher biomass, coverage, height, species diversity than south-facing slopes [49, 69, 70]. Climate warming and shifting of species towards north/ northeast in the Himalayas is reported [3, 35, 71, 72]. In Nepal, Shrestha et al. (2021) [53] projected that the target species would be elevated to 5000 m in the future, a substantial change when compared with the present distribution at 4000 m. A similar prediction for the species has been made elsewhere [3, 50].

Likewise, the change analysis, i.e., stable area, habitat loss, and habitat gain area, was carried out for RCP 4.5 and RCP 8.5 (Table 3). The area under stable habitat under RCP 4.5 (2050) was invariably higher RCP 8.5 (2050 and 2070); suggesting changes in climatic parameters and land use characteristics over higher emission rate would negatively impact the suitability of the habitat. Other than these, model prediction advocates climatic changes will also bring new areas (habitat gain) under habitat suitability class, with the highest gain observed under lower emission rate, i.e., RCP 4.5 than RCP 8.5. The study prospects the RCP4.5 scenario to be more favorable for *D. hatagirea* than RCP 8.5 in the northwestern part of IHR, thus providing an expansion scope. Meanwhile, warming of lower elevation or areas where the species is currently found explains the climatic niche loss in the future. The result obtained by overlaying the projected layer in the current and future climatic scenarios supports the projection made by Rana et al. (2021) [3] in the Nepal Himalaya using ensemble modelling (eSDM) approaches. Shreshta et al. (2021) [53] proposed a contrasting viewpoint with HadGEM2-ES, CCSM4, and BCC-CSM1-1 simulation model in a different study. Of the three models, BCC-CSM1-1 and CCSM4 simulations anticipated the species to lose 61–85% and 71–98% of its niches by 2070, whereas, with HadGEM2-ES, the species will lose all preferred niches by 2070 in RCP4.5 and RCP8.5 scenarios. In such conditions, parallel studies on community structure, biotic interactions, plant-pollinator or plant-mycorrhizal fungi interactions, population status, and habitat characteristics are also necessary. Hence, an integrated model with both the elements (biotic and abiotic) and their cumulative effects needs further investigation. This is notable because they play an important role in the persistence of *D. hatagirea* populations [52].

Along with climate change, irrational harvesting practices of MAPs as a result of increased market demand for herbal medicine and lack of knowledge/ awareness on sustainable harvesting led to serious habitat degradations in the Himalayan region [49]. Similarly, changes in LULC, anthropogenic interference (i.e., habitat degradation and fragmentation, unscientific collection, pre-mature tuber overharvesting, grazing), interspecies competition (i.e., mainly habitat engulfed by *Persicaria wallichii*), and poor seed germination has seriously impacted the relative distribution of *D. hatagirea* in the study area [49]. The present study provides an overview of habitat suitability and likely changes with projected climate scenarios in conjunction with changes in the species geomorphologic profile. In this context, the information provided in the present study could be beneficial in various conservation initiatives at the local and regional levels in West Himalaya. The study conducted has some limitations as the model developed to forecast the fundamental niche of the species rather than the realized niche. The species realized niche might not be the same as predicted in our model prediction results. Another limitation is that this study modeled the habitat suitability of *D. hatagirea* at different time scales (2050 & 2070) based on abiotic factors but did not consider biotic factors such as plant-mycorrhizal association or plant-pollinator interaction. As a result, potentially suitable area of the species might be overestimated, as biotic interaction, especially mycorrhizal association, play a crucial role in plant establishment in the family Orchidaceae [53, 73].

## 5. Conclusion

*Dactylorhiza hatagirea*, an endemic high-value Himalayan medicinal species, is under great stress and needs urgent attention. The perturbance of climate change has added extra pressure on the species allied with high degrees of anthropogenic stress. Therefore, the present study provided a clear overview of the habitat suitability for *D. hatagirea* and predicted the potential impacts of future climate on their distribution in Western Himalaya. The species potential distributions are explained mainly by elevation, precipitation of the coldest quarter, soil moisture, LULC, and mean annual temperature. The study reveals that *D. hatagirea* has approximately about 131 sq. km as its fundamental niche with high (>0.75) habitat suitability, which corroborates to about 0.24% of the total geographic area of the state. Under future projections RCP 4.5 and RCP 8.5, species distribution is projected to be very similar to the current distribution, except shifting species in a northward direction to higher elevation is expected. Furthermore, most predicted potential habitats fall within areas with anthropogenic encroachment leading to habitat degradation, unscientific and destructive harvesting practices, grazing, etc. Considering these, it is anticipated that the species may lose much of the habitat due to anthropogenic activities, while climate change impact will invariably be more at comparatively lower altitudes. Thus, it is imperative to undertake appropriate conservation steps and interventions like reintroduction/ augmentation programs. Besides these, the dependent communities awareness and sensitization curriculum holds utmost importance towards sustainable utilization. The model generated suitability maps can be of significant help to various policy prescribing National agencies, i.e., National Medicinal Plant Board (NMPB), Ministry of Environment Forest and Climate Change (MOEF &CC), Department of Science & Technology (DST). At the State level, the State Government has control over Forest Department and NGOs, thus ensuring and safeguarding the overall health of our forests and meadows.

## Supporting information

**S1 Table. Multi-collinearity test by using cross-correlations (Pearson correlation coefficients, r) among environmental variables using ENM Tools.**
(DOCX)

**S1 Fig. Habitat transformations with respect to present distribution into different classes as depicted under RCP 4.5 (2050 and 2070).** Maps in S1 Fig are generated with ArcGIS version 10.3 (ESRI, CA, USA).
(TIF)

**S2 Fig. A comparative studies on habitat transformations between present distribution and under RCP 8.5 (2050 and 2070).** Maps in S2 Fig are generated with ArcGIS version 10.3 (ESRI, CA, USA).
(TIF)

## Acknowledgments

The authors are thankful to the Director of the Institute for providing necessary facilities and encouragement. Due acknowledgment to National Remote Sensing Center, Government of India (http://www.nrsc.gov.in), WorldClim-Global Climate Data (http://www.worldclim.com) and ESA CCI Land Cover project (http://www.esa-landcover-cci.org/) for open (academic use) data source.

## Author Contributions

**Conceptualization:** Indra D. Bhatt.

**Investigation:** Laxman Singh, Nidhi Kanwar.

**Methodology:** Laxman Singh, Nidhi Kanwar.

**Project administration:** Indra D. Bhatt.

**Software:** Laxman Singh, Nidhi Kanwar.

**Supervision:** Indra D. Bhatt, Shyamal K. Nandi.

**Writing – original draft:** Laxman Singh, Nidhi Kanwar.

**Writing – review & editing:** Indra D. Bhatt, Shyamal K. Nandi, Anil K. Bisht.

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
