## [Decision Letter · Decision Letter 0]

2 Jul 2021

PONE-D-21-16347

Predicting the potential distribution of Dactylorhiza hatagirea (D. Don) Soo – a critically endangered medicinal orchid under multiple climate change scenarios

PLOS ONE

Dear Dr. Bhatt,

Thank you for submitting your manuscript to PLOS ONE. After careful consideration, we feel that it has merit but does not fully meet PLOS ONE’s publication criteria as it currently stands. Therefore, we invite you to submit a revised version of the manuscript that addresses the points raised during the review process.

We look forward to receiving your revised manuscript.

Kind regards,

Daniel de Paiva Silva, Ph.D.

Academic Editor

PLOS ONE

Journal Requirements:

"Partial financial support from NMSHE TF-3 (DST, Govt. of India) and USCS&T (Govt. of Uttarakhand) is gratefully acknowledged."

"No funding available of this work"

Additional Editor Comments (if provided):

Dr Bhatt et al.,

after two independent reviews, I believe your study may be accepted for publication in PLoS One after a significant improvement is performed to it. You will see that the reviewers had very contrasting, with one of them deciding for its rejection (but potential acceptance after deeply improved), whereas the other decided for a minor review. Considering this, I believe you should consider the issues raised by both of them, especially considering the fact that the distribution of the species was already discussed in a previous 2021 paper.

In case you decide to redo your MS, I suggest you to take special attention to the issues raised by reviewer #2.

Given all the required changes you will need to do, I will grant you a three-months period to perform the improvements (October 1st, 2021). In case you need more time, please let me know. Do not hesitate to resubmit earlier in case you are able to. In By the time you resubmit, please do not forget to prepare a rebuttal letter to inform to your reviewers of all the changes you accepted and implemented and those you did not agree with and with.

Sincerely,

Daniel Silva, Ph.D.

Reviewers' comments:

Reviewer's Responses to Questions

**Comments to the Author**

1. Is the manuscript technically sound, and do the data support the conclusions?

Reviewer #1: Partly

Reviewer #2: Yes

2. Has the statistical analysis been performed appropriately and rigorously? 

Reviewer #1: Yes

Reviewer #2: Yes

3. Have the authors made all data underlying the findings in their manuscript fully available?

Reviewer #1: Yes

Reviewer #2: Yes

4. Is the manuscript presented in an intelligible fashion and written in standard English?

Reviewer #1: Yes

Reviewer #2: Yes

5. Review Comments to the Author

Reviewer #1: Comments to the Author:

Singh et al. have produced a study to discern the potential distribution of the orchid Dactylorhiza hantagirea in the Indian state of Uttarakhand. Using the common maximum entrophy algorithm implemented in the popular software Maxent, they are building a model with several climatic, edaphic and landscape variables to produce a map with the suitable areas for this species, that is further projected to two scenarios of climate change (moderate and extreme warming). Although the authors use a standard methodology, and the models seem to be well constructed, the study has a too narrow scope. It is hard to understand why the authors have limited the niche models to just a very small range of this species (Uttarakhand), as it is actually distributed not only to the Indian Himalayas (as the authors state in lines 91-92), but on a large strip from Pakistan to Mongolia and NE China (see e.g. http://powo.science.kew.org/taxon/urn:lsid:ipni.org:names:626614-1, or http://www.efloras.org/florataxon.aspx?flora_id=2&taxon_id=242421813). The small study area, in addition to largely restrict the focus of the ms., also poses a methodological limitation to the built models: the potential occurrence in Uttarakhand is based on the fundamental niche derived from just 30 occurrences gathered by the authors. This could produce a biased model (and probably with less suitable area) if compared with a model produced with all species occurrences along its whole range. In other words, niche models, even when are produced for a small region, should be built ideally using all species’ occurrences.

Another large problem is that a very similar paper has been just published in Journal of Sustainable Forestry (https://doi.org/10.1080/10549811.2021.1923530), also using Maxent and focused in the same state (Uttarakhand). In addition, a second paper, also of 2021 and published in Journal of Applied Research on Medicinal and Aromatic Plants (https://doi.org/10.1016/j.jarmap.2020.100286), is also using Maxent to build a model for almost the whole species’ range (so, including Uttarakhand). Thus, I cannot recommend this ms. to be accepted in PLOS ONE. However, tacking the advantage that climate change models provided in the ms., and perhaps expanding its scope to more-focused conservation issues (e.g. to check whether the suitable areas fall within the present PAs in Uttarakhand), the paper could be published in a local journal or in a conservation journal.

A few specific comments:

1. Lines 89-93. The first time that the study species is mentioned, complete information about its distribution area should be provided. As I mentioned above, the species has a quite large distribution area. Some basic information about the morphology, taxonomy, ecology accross its range, and conservation status is also mandatory.

2. Lines 100-101. The authors should provide the reasons why the study area is restricted to Uttarakhand. As I mentioed above, including aims related to the local conservation issues in this state could make this paper stronger.

3. Lines 149-150. A way to improve the paper could be to incude other GCMs (such as MIROC and others).

Reviewer #2: I have read this paper with great interest. However, in its present form, there are some issues that would deserve some clarification before the paper is suitable for publication.

L36-38 I suggest removing all the Bio, and putting directly the variable name that is in parentheses.

L43-44 All these regions mentioned here should appear on the map in Figure 1. For readers, like me, who do not know the study region, they will not be able to identify and understand the results without this information in the figure.

L44-46 Considering that the title of the paper states that the study is about the evaluation of an endangered medicinal species, the conclusion of the paper should have this approach. From the title of the paper, the readers will look in the abstract for the conclusion of the study - what was found looking forward? The current conclusion is quite vague.

L48-49 These keywords (Dactylorhiza hatagirea, Habitatdistribution [space], Climate change) are already in the title. It is best to use different words

L52 Climate is an…

L63 Remove or inform which are the “etc.”

L71 Remove or inform which are the “etc.”

L74-75 I consider that before this paragraph, another paragraph with the main approach of the current manuscript needs to be developed. What are the future predictions for mountainous areas in the region? Mountain environments are quite sensitive to climate change, and small changes are already enough to alter the habitat suitability of species. I think the authors need to address this a bit, as well as changes in environmental conditions from climate change across elevational gradients. What are the expected impacts, and already known to science? What are the predicted or potential impacts on the species niche if changes occur? See this manuscript for a good approach of this: https://doi.org/10.1016/j.ecolind.2020.106435.

L100-104 Please add question marks to the questions.

L113-116 What is the range of altitude?

L258-262 Again, it is not possible to identify these regions in this figure.

L328-331 All this information is already available in the results. It is repetitive.

L421 The title of some REF are written in uppercase and others in lowercase. Standardize all of them to lowercase. Also, some REFs (4, 10, 43) are missing spaces between their names. Review REF 30.

I suggest that the authors revise the lack of space between words, because this occurred throughout the text (e.g., L41, L346, L354,....)

As the maps provided have low resolution, it was not possible to visualize the change of patterns for the different scenarios according to the changes in coloration. Anyway, besides increasing the font size and resolution of these figures, I suggest that the authors put the name of the axis with the biological variable (e.g., minimum temperature), and not the code given by WorldClim (e.g., Bio 2). But this code could come in parentheses.

6. PLOS authors have the option to publish the peer review history of their article (what does this mean?). If published, this will include your full peer review and any attached files.

Reviewer #1: No

Reviewer #2: No

---

## [Author Response · Author response to Decision Letter 0]

8 Nov 2021

Reviewer 1

1. Although the authors use a standard methodology, and the models seem to be well constructed, the study has a too narrow scope. It is hard to understand why the authors have limited the niche models to just a very small range of this species (Uttarakhand), as it is actually distributed not only to the Indian Himalayas (as the authors state in lines 91-92), but on a large strip from Pakistan to Mongolia and NE China 

Response: As indicated by the reviewer, some insights on the potential distribution of the target species across the Himalayan region was given by Thakur et al. (2021), however, detailed investigation at the regional level was identified as gap area of research. Resource limitation to validate the model in other part of the Himalayan region, asserts the present study to the state of Uttarakhand. Further, on the realistic side, we have extensively surveyed the state of Uttarakhand for population estimation along the altitudinal gradient and in different habitat types [Singh et al. (2021); doi doi.org/10.1007/s10113-021-01762-6] and validated the model output with the field data. Needful revision on the manuscript, on the distribution of the target species is incorporated and revised.

2. Another large problem is that a very similar paper has been just published 

Response: We believe that every study conducted has some merits and limitations. As indicated, previously two papers are published. In a study by Thakur et al. (2021), emphasized on the ecological factors affecting the occurrence of the species and ground truth for Himachal Pradesh (India) only, although they have model the presence of the species for the entire Himalayan region. In another study by Chandra et al. (2021), their work was concentrated specifically in the Uttarakhand state and ground truth. Likely impact of climate change on the species suitable habitat and transformation thereafter was the gap area of research. Thus, taking the learning’s from both the study, the present study envisaged to identify region/pockets where the target species habitat suitability was maximum, moderate, low or least. Further, we intended to identify, how the present distribution would be affected in the face of climate change using different climate projections parameters. Also, habitat transformation (i.e., stable, gain or loss) and change analysis was conducted to depict vulnerable areas and areas which will remain suitable in the next 50 years

3. Lines 89-93. The first time that the study species is mentioned, complete information about its distribution area should be provided. As I mentioned above, the species has a quite large distribution area. Some basic information about the morphology, taxonomy, ecology across its range, and conservation status is also mandatory. 

Response: We thank the reviewer for the suggestion. The necessary information asked for is added into the manuscript.

4. Lines 100-101. The authors should provide the reasons why the study area is restricted to Uttarakhand. As I mentioned above, including aims related to the local conservation issues in this state could make this paper stronger. 

Response: We agree with the reviewer viewpoint regarding elsewhere reported records of the target species. The state of Uttarakhand was extensively ground truth, which in others cases would not have been possible. Further, without validating a model with the field data would be biased in true sense. 

5. Lines 149-150. A way to improve the paper could be to incude other GCMs (such as MIROC and others). 

Response: We thank the reviewer for his/her remark. After extensive literature review on Himalayan species distribution modelling, the present model was selected. We believe there is prediction disparity among different models in different regions. 

Reviewer 2:

1 L36-38 I suggest removing all the Bio, and putting directly the variable name that is in parentheses. 

Response: We thank the reviewer for bringing up this correction. Needful actions has been taken in the revised manuscript

2 L43-44 All these regions mentioned here should appear on the map in Figure 1. For readers, like me, who do not know the study region, they will not be able to identify and understand the results without this information in the figure. 

Response: Map of the Figure 1 has been revised extensively, with the region having high habitat suitability marked.

3 L44-46 Considering that the title of the paper states that the study is about the evaluation of an endangered medicinal species, the conclusion of the paper should have this approach. From the title of the paper, the readers will look in the abstract for the conclusion of the study - what was found looking forward? The current conclusion is quite vague. 

Response: We thank the reviewer for his/her constructive remark. In the revised manuscript, abstract conclusion has been made more conclusive and drawn factual statement.

4 L48-49 These keywords (Dactylorhiza hatagirea, Habitat distribution [space], Climate change) are already in the title. It is best to use different words 

Response: We thank the reviewer for his/her suggestions. Necessary actions has been undertaken and revised 

5 L52 Climate is an… 

Response: Incorporated as suggested

6 L63 Remove or inform which are the “etc.” 

Response: In the revised manuscript “etc” has been removed as suggested

7 L71 Remove or inform which are the “etc.” 

Response: In the revised manuscript “etc” has been removed as suggested

8 L74-75 I consider that before this paragraph, another paragraph with the main approach of the current manuscript needs to be developed. What are the future predictions for mountainous areas in the region? Mountain environments are quite sensitive to climate change, and small changes are already enough to alter the habitat suitability of species. I think the authors need to address this a bit, as well as changes in environmental conditions from climate change across elevational gradients. What are the expected impacts, and already known to science? What are the predicted or potential impacts on the species niche if changes occur? 

Response: We thank the reviewer for the suggestions. In the revised manuscript, a separate paragraph has been added to address all the suggestion raised by the reviewer.

9 L100-104 Please add question marks to the questions. 

Response: Necessary action is undertaken

10 L113-116 What is the range of altitude? 

Response: In the Figure 1, altitudinal class across different region is provided, so as to depict highland and lowland areas within the state boundary

11 L258-262 Again, it is not possible to identify these regions in this figure. 

Response: The regions mentioned in the manuscript has been outlined with respective boundary files and respectively labeled

12 L328-331 All this information is already available in the results. It is repetitive. 

Response: The repetitive text has been deleted

13 L421 The title of some REF are written in uppercase and others in lowercase. Standardize all of them to lowercase. Also, some REFs (4, 10, 43) are missing spaces between their names. Review REF 30. 

Response: Necessary actions has been undertaken and revised

14 I suggest that the authors revise the lack of space between words, because this occurred throughout the text (e.g., L41, L346, L354,....) 

Response: Necessary actions has been undertaken and revised

15 As the maps provided have low resolution, it was not possible to visualize the change of patterns for the different scenarios according to the changes in coloration. 

Response: We support the reviewer viewpoint in the revised manuscript; the resolution of the maps has been increased. Also, change-shift maps has been divided into separate maps of each, so as to make them more visible and meaningful

16 Anyway, besides increasing the font size and resolution of these figures, I suggest that the authors put the name of the axis with the biological variable (e.g., minimum temperature), and not the code given by WorldClim (e.g., Bio 2). But this code could come in parentheses. 

Response: We thank the reviewer for the suggestion, the highlighted figures has been revised.

---

## [Decision Letter · Decision Letter 1]

13 Jan 2022

PONE-D-21-16347R1Predicting the potential distribution of Dactylorhiza hatagirea (D. Don) Soo – a critically endangered medicinal orchid under multiple climate change scenariosPLOS ONE

Dear Dr. Bhatt,

Thank you for submitting your manuscript to PLOS ONE. After careful consideration, we feel that it has merit but does not fully meet PLOS ONE’s publication criteria as it currently stands. Therefore, we invite you to submit a revised version of the manuscript that addresses the points raised during the review process.

We look forward to receiving your revised manuscript.

Kind regards,

Daniel de Paiva Silva, Ph.D.

Academic Editor

PLOS ONE

Additional Editor Comments:

Dear Bhatt et al.,

In light of the reviews provided by both reviewers, I find your manuscript could be accepted to be published in PLoS One after the passes through a major review. Please resubmit until April 12, 2021. If you need more time, please let me know and do not hesitate in case you are able to resubmit earlier. Please take close attention to all the issues raised by both reviewers. Unfortunately, there are several points that you need to improve you manuscript yet.

By the time of resubmission, please do not forget to prepare a rebuttal letter for your reviewers, explaining each and every decision you made during the review process.

Sincerely,

Daniel Silva

Reviewers' comments:

Reviewer's Responses to Questions

**Comments to the Author**

1. If the authors have adequately addressed your comments raised in a previous round of review and you feel that this manuscript is now acceptable for publication, you may indicate that here to bypass the “Comments to the Author” section, enter your conflict of interest statement in the “Confidential to Editor” section, and submit your "Accept" recommendation.

Reviewer #3: (No Response)

Reviewer #4: (No Response)

2. Is the manuscript technically sound, and do the data support the conclusions?

Reviewer #3: Partly

Reviewer #4: Yes

3. Has the statistical analysis been performed appropriately and rigorously? 

Reviewer #3: Yes

Reviewer #4: No

4. Have the authors made all data underlying the findings in their manuscript fully available?

Reviewer #3: Yes

Reviewer #4: Yes

5. Is the manuscript presented in an intelligible fashion and written in standard English?

Reviewer #3: No

Reviewer #4: No

6. Review Comments to the Author

Reviewer #3: In their study, the authors predicted the potential distribution (current and future) of a medicinal orchid, Dactylorhiza hatagirea, which becomes even rarer due to the intense human activity in the Himalayan area. Specifically, they used Maxent algorithm to predict the potential distribution of D. hatagirea, a software classified among those with the highest predictive accuracy. From a methodological perspective, I disagree with the use of AUC value as a measure of model evaluation. Although this metric has been widely used so far, I believe that others are more important (e.g. AICc; see specific comment below). In general, authors did not use the findings of Thakur et al. (2021) and Shrestha et al. (2021) who also predicted the current and future potential distribution in the Himalayan are (India and Nepal). In the light of these two existing studies, the present study should highlight the novelty of their own study.

However, I believe that the most significant problem in the language! Although I am not a native English speaker, I feel that the manuscript would be greatly improved if it would be checked by someone with excellent skills in the English language.

Others suggestions

P.3, L.56: “ ……. and abiotic factor affecting species potential geographic…..”

P.3, L.71: “….. is much higher than the global average”. Delete the word “rate”.

P.4, L. 83: “….. with 31% of them being native, whereas 15.5 % endemic and threatened.”

P.4, L. 85: “Other challenges …..”

P.4, L.97: “Therefore, estimation current plants distribution and identifying important climatic refugia will help predict future distribution …..”

P.4, L.99: ecological niche modelling is a set of different techniques, whereas Maxent is just one of them. So, these two terms should not be treated as the same. In your case I would suggest to make a small introduction to the SDMs in general, and afterwards you could write one or two sentences about Maxent.

P.5, L. 101. “Using these input variables”. Which variables? You did not mention any kind of variables. You should say “Using environmental variables/predictors …”.

P5-6, L.105-127: I suggest stating other authors (Thakur et al. 2021; Shrestha et al. 2021) who predicted current and future potential distribution of D. hatagirea and after that you can add a part about the novelty of your study.

P.6, L.140: “…. harbors alpine vegetation, which covers …..”

P.6, L. 141: Replace the phrase “in totality it forms about” with “corresponding to c. 24.11%”

P.6, L.142: Either write “The alpine areas are well known for their high-value …” or “The alpine area is well known for its high-value ….”

P.7, L. 147: “presence points” instead of “presence point”

P7, L.150-151: “Data on the species were very limited, whereas herbarium records were not geo-referenced.”

P.7, L. 152: “presence points were recorded.”

P.7, L.155: “decimal degrees” instead of “degrees decimal”

P.8, L.168: can you please say a few words why you used CCSM4 instead of others?

P.8, L. 172: “edaphic properties are expected to remain stable”

P.8, L. 173: “hence the same raster layer was used in future projections.”

P.9, L.178-189. This part should be rewritten and should be checked by a native English speaker.

P.10, L. 232. I wouldn’t say that reduction in the number of predictors increases the predictive power of the model. Instead of that, if you use highly inter-correlated variables, then you will not be able to identify those that are highly important for the distribution of the studied species. And this is owned to the high correlation coefficient between different predictors.

Authors used AUC to assess the predictive accuracy of their model. Actually, I didn’t read that they used the average model prediction made after 30 runs. However, high AUC values, as in the case of this study, may be owned to the low number of species records. Instead of using AUC value, I would recommend of running the model a number of times (e.g. 10 or 30 runs) and then select the best model by using the Akaike information criterion (AICc) (Warren & Seifert 2011).

P. 13. L. 272: “The TIC value is the habitats suitability class of D. hatagirea on the maxent model.” What do you mean? Please rephrase that sentence.

In several parts of the manuscript, a semi-colon (;) was used instead of a comma.

p.17, L. 337: Authors used the term “Habitat distribution modelling”. This term is used here for the first time, instead of others that were used in other parts of the manuscript. I would suggest using the same terminology throughout the text.

p. 17, L.340: “under a changing climate scenario”

p.17, L.341-342: SDMs provide the potential distribution. However, species distributions are determined both by abiotic factors and biotic interactions. Orchids are characterized by strong biotic interactions which have to do with the mycorrhizal fungi that will help them germinate and keep feeding them, as well as by specific insects which will be their pollinators. Such biotic interactions are very important and, in many cases, can influence orchids’ potential distribution. Such limitations should be mentioned in the manuscript. You can read Evans et al (2021) and Tsiftsis & Djordjevic (2020).

P.18, L.349-350: I think that the term “species density” is wrong. Can you rewrite that part as this term is referring to a number of different species?

P.18, L. 354: add a comma after the word “variables”.

p.18, L.355-358: please rewrite that part. It is not clear what you want to say!

The findings of this study should be discussed in relation to the findings of Thakur et al. (2021) and Shrestha et al. (2021), which at this stage are ignored.

References

Evans, A., Janssens, S. & Jacquemyn, H. 2021. Impact of Climate Change on the Distribution of Four Closely Related Orchis (Orchidaceae) Species. Diversity 12(8): 312. DOI: 10.3390/d12080312

Shrestha, B., Tsiftsis, S., Chapagain, D.-J., Khadka, C., Bhattarai, P., Kayastha, N., Kolanowska, M. & Kindlmann, P. 2021. Dactylorhiza hatagirea in Nepal: Distribution prediction under current and future climate change context. Plants 10(3):467.

Tsiftsis, S. & Djordjević, V. 2020. Modelling sexually deceptive orchid species distributions under future climates: the importance of plant-pollinator interactions. Scientific Reports 10, 10623.

Warren, D. L., & Seifert, S. N. (2011). Ecological niche modeling in Maxent: the importance of model complexity and the performance of model selection criteria. Ecological Applications, 21, 335–342.

Reviewer #4: Review of the manuscript “Prediction the potential distribution of Dactylorhiza hatagirea (D. Don) Soo – a critically endangered medicinal orchid under multiple climate change scenarios” by Singh et al.

In this study, Singh et al aims to identify habitat suitability of D. hatagirea in the Western Himalaya and determine the future geographical distribution under climate change scenarios.

Looking at the comments and suggestions from two previous reviewers and responses from authors, I see authors include suggestions from reviewers and I believe the manuscript has improve substantially compared to the previous version. However, I have some concerns on how analysis were conducted. I have some suggestions/comments which may help authors for a more clear analysis.

Abstract

L32. With these models, the authors are not determining the future geographical distribution, but the suitable areas (in terms of environmental variables).

L41. Most of the species loss habitat suitability in future scenarios. Especially when they move to higher altitude which by the conic shape of mountains, area is less available. Of course this is possible if species may disperse to different areas where is found in contemporary habitat.

Add in what percentage is expected to expand.

Introduction

L59. Not necessarily. Some species are more capable than others, not all species will show the same responses.

L69. The increased in greenhouse gases emission is not only for the region, but for the entire planet.

L106. The species is an orchid, right? Authors should say this here.

L116. Not clear what do the authors mean with “the species require…..conditions for growth and perturbation….”

L118. “The existing multiplication for mass multiplication” sounds odd to me.

L121. Delete “present”

Materials and methods

I´d suggest a section for Study species with a brief taxonomic description and more on the use and management. I´d include detailed pictures of the species (flowers, leaves, bulbs), individuals in the wild and also pictures of the products in trade.

Is the species included in the IUCN Red List? In case is not I´d highly recommend to make the IUCN assessment following the IUCN criteria and include this as part of results.

2.3. Climate data. I suggest a more general subtitle for this subsection. What the authors are using is not only climate data. They are using geomorphologic, pedologic and LULC.

The land use is an anthropogenic layer and as used, I believe is a categorical variable. This layer should not be included as a predictor variable in the model. Instead the resulting model can be clipped with this layer only as a polygon.

L212. AUC was the only statistical test used assess the models?

Since several years ago, AUC has been demonstrated its reliability as a comparative measure of accuracy. I suggest to include some other tests, (for example the binomial test, Partial ROC, Trus skill statistics). Lobo et al. 2008 (https://doi.org/10.1111/j.1466-8238.2007.00358.x) is a must read article for this.

L217. The used classification especially for comparison among RCPs and periods (Table 4; Figure 5, Figure 7 and 8) is quite confusing at least to me. I would use instead a binary classification (0/1, suitable/unsuitable) using only one threshold. This way would be much more clear to compare among RCPs and periods (see suggestions in the Results section).

Results

L235. Table 1 is in the Methods section, but it is a Result. I suggest to include it only here and not in Methods.

Table 1. I see the information on this table as a result. Why is it in methods?

Table 3. It is not clear to me how the Gain, Loss, Stable and Unsuitable areas were computed. Authors did use a classification of Unsuitalbe, low suitable, moderate suitable

Table 4. Are the authors presenting the same information in Table 4 than in the Maps (figure 7 and 8)? Highly recommended to include the information only once.

Figures

Fig. 1. I suggest to add (at least in a small inset) the Map of India.

Figures 2-4 are those generated by default by Maxent. I don´t see these figures essential to be included in the results. They could not be included or maybe only as supplementary.

Figures 7 and 8 and the colors used are not useful to explore what is the future of the species. The main problem from my point of view is the 11 categories resulting of comparisons between RCPs and years (2050 and 2070). I would not use this comparasion. I would do the comparison using the gain, loss, stable and unsuitable categories, considering only the binary classification.

7. PLOS authors have the option to publish the peer review history of their article (what does this mean?). If published, this will include your full peer review and any attached files.

Reviewer #3: No

Reviewer #4: No

---

## [Author Response · Author response to Decision Letter 1]

4 Apr 2022

1 I believe that the most significant problem in the language! 

Although I am not a native English speaker, I feel that the manuscript would be greatly improved if it would be checked by someone with excellent skills in the English language. We thank the reviewer for the needful revision; in the revised manuscript, grammatical errors and other languages shortfalls has been significantly improved

2 P.3, L.56: “ ……. and abiotic factor affecting species potential geographic…..” Revised as suggested (L. 56-57)

3 P.3, L.71: “….. is much higher than the global average”. Delete the word “rate”. We thank the reviewer for the suggestion. Needful revisionsareundertakenin the manuscript (L. 71)

4 P.4, L. 83: “….. with 31% of them being native, whereas 15.5 % endemic and threatened.” Revised as suggested (L. 82-83)

5 P.4, L. 85: “Other challenges …..” We thank the reviewer for his/her remark. Needful revision is made in the manuscript (L. 84)

6 P.4, L.97: “Therefore, estimation current plants distribution and identifying important climatic refugia will help predict future distribution …..” Needful revision is undertaken in the revised manuscript (L. 96)

7 P.4, L.99: ecological niche modelling is a set of different techniques, whereas Maxent is just one of them. So, these two terms should not be treated as the same. In your case I would suggest to make a small introduction to the SDMs in general, and afterwards you could write one or two sentences about Maxent. We thank the reviewer for the needful correction. The revised manuscript differentiates between MaxEnt and other simulation tools. Also, as suggested, a small introduction about SDM is added.

(L. 99-104)

8 P.5, L. 101. “Using these input variables”. Which variables? You did not mention any kind of variables. You should say “Using environmental variables/predictors …”. We thank the reviewer for his/her remark.

Needful corrections are undertaken

(L. 104-105)

9 P5-6, L.105-127: I suggest stating other authors (Thakur et al. 2021; Shrestha et al. 2021) who predicted current and future potential distribution of D. hatagirea and after that you can add a part about the novelty of your study. With reference to Comment No. 31 (Sl. No), we have briefly discussed the finding of both authors and integrated them with the present study findings

(L. 352-472)

10 P.6, L.140: “…. harbors alpine vegetation, which covers …..” Revised as suggested (L. 145)

11 P.6, L. 141: Replace the phrase “in totality it forms about” with “corresponding to c. 24.11%” Revised as suggested (L. 146)

12 P.6, L.142: Either write “The alpine areas are well known for their high-value …” or “The alpine area is well known for its high-value ….” Revised as suggested (L. 147)

13 P.7, L. 147: “presence points” instead of “presence point” We thank the reviewer for the remark; needful correction is made in the revised manuscript (L. 152)

14 P7, L.150-151: “Data on the species were very limited, whereas herbarium records were not geo-referenced.” Needful correction is made

(L. 155-156)

15 P.7, L. 152: “presence points were recorded.” Revised as suggested (L. 157)

16 P.7, L.155: “decimal degrees” instead of “degrees decimal” We thank the reviewer for his/her remark; needful revision is made in the revised manuscript (L. 160)

17 P.8, L.168: can you please say a few words why you used CCSM4 instead of others? After extensive literature review and synthesis from the studies (mainly from the Hindu-Kush Himalayan region), we needed a balance model to fulfill our objective

18 P.8, L. 172: “edaphic properties are expected to remain stable” Revised as suggested (L. 176-177)

19 P.8, L. 173: “hence the same raster layer was used in future projections.” Revised as suggested (L. 178)

20 P.9, L.178-189. This part should be rewritten and should be checked by a native English speaker. We thank the reviewer for the needful revision; this section has been revised to develop a meaningful sentence.

(L. 182-191)

21 P.10, L. 232. I wouldn’t say that reduction in the number of predictors increases the predictive power of the model. Instead of that, if you use highly inter-correlated variables, then you will not be able to identify those that are highly important for the distribution of the studied species. And this is owned to the high correlation coefficient between different predictors. We thank the reviewer for the suggestion, and we are in line with/ agree with the reviewer's viewpoints. The sentence in the revised manuscript has beenrephrased

(L. 243-246)

22 Authors used AUC to assess the predictive accuracy of their model. Actually, I didn’t read that they used the average model prediction made after 30 runs. However, high AUC values, as in the case of this study, may be owned to the low number of species records. Instead of using AUC value, I would recommend of running the model a number of times (e.g. 10 or 30 runs) and then select the best model by using the Akaike information criterion (AICc) (Warren & Seifert 2011). We thank the reviewer for his/her remark. The needful revision sought is revised.

In the revised manuscript, other than relying on AUC for assessing the model accuracy, we also calculated sensitivity, specificity, and true skill statistics (TSS) (Lobo et al. 2008; https://doi.org/10.1111/j.1466-8238.2007.00358.x), which are widely used to access the model accuracy. 

(L. 220-229 & L. 254-259)

23 P. 13. L. 272: “The TIC value is the habitats suitability class of D. hatagirea on the maxent model.” What do you mean? Please rephrase that sentence. We thank the reviewer for the needful revision; the sentence has been deleted as the convictionmade failed to make any fruitful argument in the preceding paragraph (L. 289-90)

24 In several parts of the manuscript, a semi-colon (;) was used instead of a comma. The manuscript has been thoroughly revised with all the parameters in consideration of the present comments

25 p.17, L. 337: Authors used the term “Habitat distribution modelling”. This term is used here for the first time, instead of others that were used in other parts of the manuscript. I would suggest using the same terminology throughout the text We thank the reviewer for his/her remark. In the revised manuscript, for the shake of uniformity, SDM is used throughout the manuscript

26 p. 17, L.340: “under a changing climate scenario” Revised as suggested (L. 360)

27 p.17, L.341-342: SDMs provide the potential distribution. However, species distributions are determined both by abiotic factors and biotic interactions. Orchids are characterized by strong biotic interactions which have to do with the mycorrhizal fungi that will help them germinate and keep feeding them, as well as by specific insects which will be their pollinators. Such biotic interactions are very important and, in many cases, can influence orchids’ potential distribution. Such limitations should be mentioned in the manuscript. You can read Evans et al (2021) and Tsiftsis&Djordjevic (2020). We thank the reviewer for the needful revision. The needful suggestion has been incorporated, and limitations of the study are adequately mentioned

(L. 466-473)

28 P.18, L.349-350: I think that the term “species density” is wrong. Can you rewrite that part as this term is referring to a number of different species? We thank the reviewer for the suggestion; needful action in the sentence is undertaken in the revised manuscript

(L. 409-412)

29 P.18, L. 354: add a comma after the word “variables”. Revised as suggested

30 p.18, L.355-358: please rewrite that part. It is not clear what you want to say! We thank the reviewer for the needful revision. This section has been re-written and revised accordingly (L. 409-417)

31 The findings of this study should be discussed in relation to the findings of Thakur et al. (2021) and Shrestha et al. (2021), which at this stage are ignored. We thank the reviewer for the suggestion. In the discussion section, both the publication and other studies conducted on the target species are adequately discussed with the present study's finding. (L. 353-373)

## Reviewer- 4

S. No. Comments/suggestions/ queries Response

1 L32. With these models, the authors are not determining the future geographical distribution, but the suitable areas (in terms of environmental variables). We thank the reviewer for bringing up this correction. Needful actions havebeen undertaken and revised accordingly (L. 32-34)

2 L41. Most of the species loss habitat suitability in future scenarios. Especially when they move to higher altitude which by the conic shape of mountains, area is less available. Of course this is possible if species may disperse to different areas where is found in contemporary habitat.

Add in what percentage is expected to expand. We thank the reviewer for the suggestion; needful expansion percent has been incorporated in the revised manuscript (L. 40-42)

3 L59. Not necessarily. Some species are more capable than others, not all species will show the same responses. We partially agree with the reviewer's viewpoint. Yes, all species may not reflect the same response, but here, we intend to emphasizeon those species (especially in the context of the Himalayan region) where such responses are adequately seen. Such responses are often among the Rare, Endangered, and Threatened (RET) species in the Himalayan region. Although we still believe more conclusive needs to be conducted to strengthen this argument elsewhere and in the Himalayas.

(Manish, 2022; https://doi.org/10.1016/j.ecoinf.2021.101546)

Rather et al. (2022) (https://doi.org/10.1016/j.ecoleng.2021.106534)

Hamid et al. (2018) (https://doi.org/10.1007/s10531-018-1641-8)

4 L69. The increased in greenhouse gases emission is not only for the region, but for the entire planet. We agree with the reviewer's viewpoint, but if we compare the impacts of climate change across the globe and the Himalayasin several reports, including IPPC and several other projection reports, the cascading impacts of climate change in this region arejust double than that of the global average.

https://report.ipcc.ch/ar6wg2/pdf/IPCC_AR6_WGII_CrossChapterPaper5.pdf

Sabin et al. (2020) (https://doi.org/10.1007/978-981-15-4327-2_11)

Liu J, Rasul G. Climate change, the Himalayan mountains, and ICIMOD. Sustainable mountain development. 2007; 53:11-4.

Rafiq et al. (2022) (https://doi.org/10.1007/978-3-030-89308-8_12)

5 L106. The species is an orchid, right? Authors should say this here. For kind consideration and also for better understanding of the readers, the identity of the species is added in the Title of the manuscript itself (L. 1-2; L. 29)

6 L116. Not clear what do the authors mean with “the species require…..conditions for growth and perturbation….” Other than being an endemic (i.e., Himalaya), the target species also have high habitat preferentially towards a particular set of conditions. In this region, local microclimatic conditions play an important role in shaping or determining the extent of such species' distribution.

The sentence has been re-phrased for better understanding (L. 118-119)

7 L118. “The existing multiplication for mass multiplication” sounds odd to me. We thank the reviewer for highlighting this; the sentence has been re-phrased in the revised manuscript (L. 119-122)

8 L121. Delete “present” Revised as suggested (L. 122-123)

9 I´d suggest a section for Study species with a brief taxonomic description and more on the use and management. I´d include detailed pictures of the species (flowers, leaves, bulbs), individuals in the wild and also pictures of the products in trade. We thank the reviewer for his/her remarks. In the introduction, a brief taxonomic description of characteristics is added. (L. 110-112)

A separate Figure 1 is added in the revised manuscript to address the follow-up question. (L. 131-132)

10 Is the species included in the IUCN Red List? In case is not I´d highly recommend to make the IUCN assessment following the IUCN criteria and include this as part of results? As per IUCN 2022 (https://www.iucnredlist.org/) the species is not included in the threatened category; however, at the regional level, the species is categorized as ‘Critically Endangered’ (CAMP; http://envis.frlht.org/camp.php).

As the species is not included in IUCN 2022 threat category, the Title of the manuscript is revised (L. 1-2).

In the subsequent publication, the threat assessment as per the IUCN criteria would be undertaken

11 2.3. Climate data. I suggest a more general subtitle for this subsection. What the authors are using is not only climate data. They are using geomorphologic, pedologic and LULC. We thank the reviewer for highlighting this. A general title as ‘Data source’ is included in the revised manuscript for better understanding(L.162)

12 L212. AUC was the only statistical test used assess the models?Since several years ago, AUC has been demonstrated its reliability as a comparative measure of accuracy. I suggest to include some other tests, (for example the binomial test, Partial ROC, Trus skill statistics). Lobo et al. 2008 (https://doi.org/10.1111/j.1466-8238.2007.00358.x) is a must read article for this. We thank the reviewer for the suggestion and agree with his/her viewpoint. Following this, in the revised manuscript, other than relying on AUC for assessing the model accuracy; we also calculated sensitivity, specificity, and true skill statistics (TSS) using confusion matrix preparation according to Alloche et al. (2006) (doi: 10.1111/j.1365-2664.2006.01214.x) 

(L. 220-229 & L. 254-259)

13 L217. The used classification especially for comparison among RCPs and periods (Table 4; Figure 5, Figure 7 and 8) is quite confusing at least to me. I would use instead a binary classification (0/1, suitable/unsuitable) using only one threshold. This way would be much more clear to compare among RCPs and periods (see suggestions in the Results section). We thank the reviewer for the suggestion. As stated in Comments No. 6 (Sl. No) regarding high habitat preferentiality and contagious distribution (mostly) of the species, here we aimed to elucidate and identify pockets that are most, moderate, low, and least suitable habitat in the given study area. Doing this, it is envisaged to provide a clear-cut picture of where the species can be translocated or habitats that can be taken up for medicinal plant conservation and developmental area.

As Figures 7 & 8 giveaway the same information as given in Table 4, therefore in the revised manuscript, both the Figures are added as supplementary Figures 1 & 2. (L. 325)

The habitat suitability classification used in this study was prepared following the methodology of Adhikari et al. (2012) (doi:10.1016/j.ecoleng.2011.12.004), Yang et al. (2013)(http://dx.doi.org/10.1016/j.ecoleng.2012.12.004), and Zhang et al. (2019) (https://doi.org/10.1016/j.ecoinf.2019.01.004); 

Arslan et al. (2020) (https://doi.org/10.1007/s10113-020-01695-6)

14 L235. Table 1 is in the Methods section, but it is a Result. I suggest to include it only here and not in Methods. 

We thank the reviewer for the suggestion; needful revision is incorporated in the revised manuscript

(L. 247)

15 Table 1. I see the information on this table as a result. Why is it in methods? 

16 Table 3. It is not clear to me how the Gain, Loss, Stable and Unsuitable areas were computed. Authors did use a classification of Unsuitalbe, low suitable, moderate suitable The geographical distribution of the species in terms of unsuitable, low-moderate suitable, and high suitability was initially determined using the classification of Adhikari et al. (2012) (doi:10.1016/j.ecoleng.2011.12.004), Yang et al. (2013)(http://dx.doi.org/10.1016/j.ecoleng.2012.12.004), and Zhang et al. (2019) (https://doi.org/10.1016/j.ecoinf.2019.01.004); 

Arslan et al. (2020) (https://doi.org/10.1007/s10113-020-01695-6)

Subsequent to this, to determine gain, loss, or stable habitats, we took the pixels value of the current distribution (of the above-classified habitats) and subtracted them with the pixels values of the classified habitats of the two climatic scenarios (i.e., 2050 & 2070).

A similar study is undertaken in some of the previous studies:

Coban et al. (2010) Investigation on changes in complex

vegetation coverage using multi-temporal landsat data of WesternBlack sea region-a case study. J Environ Biol 31:169–178

Arslan et al. (2020): https://doi.org/10.1007/s10113-020-01695-6

17 Table 4. Are the authors presenting the same information in Table 4 than in the Maps (figure 7 and 8)? Highly recommended to include the information only once. We thank the reviewer for the suggestion, as Fig. 7 & Fig. 8 represent the same information as in Table 4; therefore, we decided to add both the Figures as supplementary figures in the revised manuscript

(L. 325)

18 Fig. 1. I suggest to add (at least in a small inset) the Map of India. Figure 1 has been revised as per suggestion, and now in the revised manuscript, Figure 1 is marked as Figure 2 due addition of a new Figure as Figure 1 in its place

19 Figures 2-4 are those generated by default by Maxent. I don´t see these figures essential to be included in the results. They could not be included or maybe only as supplementary. We thank the reviewer for the suggestions, but we believe retaining both the Figures might be helpful for knowledgeable readers and understanding parameters having a larger influence on the species distribution

20 Figures 7 and 8 and the colors used are not useful to explore what is the future of the species. The main problem from my point of view is the 11 categories resulting of comparisons between RCPs and years (2050 and 2070). I would not use this comparison. I would do the comparison using the gain, loss, stable and unsuitable categories, considering only the binary classification. We thank the reviewer for the suggestion, as Fig.7 and Fig.8 represent the same information as mentioned in Table 4; therefore, both the Figures are now added assupplementary datasets.

Yes, using 11 color categories can be quite confusing, but at the same time, our aim of the study was to pinpoint areas that are expected to see transformation in their habitat using the climate projection model for better advocacy of the management plan.

---

## [Decision Letter · Decision Letter 2]

26 May 2022

Predicting the potential distribution of Dactylorhiza hatagirea (D. Don) Soo-an important medicinal orchid in the West Himalaya, under multiple climate change scenarios 

PONE-D-21-16347R2

Dear Dr. Bhatt,

We’re pleased to inform you that your manuscript has been judged scientifically suitable for publication and will be formally accepted for publication once it meets all outstanding technical requirements.

Kind regards,

Daniel de Paiva Silva, Ph.D.

Academic Editor

PLOS ONE

Additional Editor Comments (optional):

Dear Dr. Bhatt,

It is a pleasure to inform you that your manuscript was formally accepted for publication in PLoS One. Congratulations!

Daniel Silva, PhD.

Reviewers' comments:

Reviewer's Responses to Questions

**Comments to the Author**

1. If the authors have adequately addressed your comments raised in a previous round of review and you feel that this manuscript is now acceptable for publication, you may indicate that here to bypass the “Comments to the Author” section, enter your conflict of interest statement in the “Confidential to Editor” section, and submit your "Accept" recommendation.

Reviewer #3: (No Response)

Reviewer #4: All comments have been addressed

2. Is the manuscript technically sound, and do the data support the conclusions?

Reviewer #3: Yes

Reviewer #4: Yes

3. Has the statistical analysis been performed appropriately and rigorously? 

Reviewer #3: Yes

Reviewer #4: Yes

4. Have the authors made all data underlying the findings in their manuscript fully available?

Reviewer #3: No

Reviewer #4: Yes

5. Is the manuscript presented in an intelligible fashion and written in standard English?

Reviewer #3: Yes

Reviewer #4: Yes

6. Review Comments to the Author

Reviewer #3: (No Response)

Reviewer #4: Thanks to authors for taking into consideration and include most of the suggestions. I believe this new version make an important contribution for the conservation and sustainable management to Dactylorhiza hatagirea. Authors have included clearly most of suggestions and those they do not, they soundly justify and explain.

7. PLOS authors have the option to publish the peer review history of their article (what does this mean?). If published, this will include your full peer review and any attached files.

Reviewer #3: No

Reviewer #4: **Yes: **Leonel Lopez Toledo

---

## [Editor Report · Acceptance letter]

10 Jun 2022

PONE-D-21-16347R2 

Predicting the potential distribution of *Dactylorhiza hatagirea* (D. Don) Soo-an important medicinal orchid in the West Himalaya, under multiple climate change scenarios 

Dear Dr. Bhatt:

I'm pleased to inform you that your manuscript has been deemed suitable for publication in PLOS ONE. Congratulations! Your manuscript is now with our production department. 

Kind regards, 

on behalf of

Dr. Daniel de Paiva Silva 

Academic Editor

PLOS ONE